# A survey for characterizing details of fall events experienced by lower limb prosthesis users

Andrew Sawers[1]☯*, Cody L. McDonald[2]‡, Brian J. Hafner[2]☯

1 Department of Kinesiology and Nutrition, College of Applied Health Sciences, University of Illinois at Chicago, Chicago, Illinois, United States of America, 2 Division of Prosthetics and Orthotics, Department of Rehabilitation Medicine, University of Washington, Seattle, Washington, United States of America

☯ These authors contributed equally to this work.
‡ CLM also contributed equally to this work.
* asawers@uic.edu

**Data Availability Statement:** All relevant data are within the paper and its Supporting information files.

**Funding:** This work was supported supported by Otto Bock Healthcare LP and Össur R&D (AS), as

## Abstract

Despite their importance to fall prevention research, little is known about the details of real-world fall events experienced by lower limb prosthesis users. This gap can be attributed to the lack of a structured, population-specific fall survey to document these adverse health events. The objective of this project was to develop a survey capable of characterizing the circumstances and consequences of fall events in lower limb prosthesis users. Best practices in survey development, including focus groups and cognitive interviews with diverse samples of lower limb prosthesis users, were used to solicit input and feedback from target respondents, so survey content would be meaningful, clear, and applicable to lower limb prosthesis users. Focus group data were used to develop fall event definitions and construct a conceptual fall framework that guided the creation of potential survey questions and response options. Survey questions focused on the activity, surroundings, situation, mechanics, and consequences of fall events. Cognitive interviews revealed that with minor revisions, survey definitions, questions, and response options were clear, comprehensive, and applicable to the experiences of lower limb prosthesis users. Administration of the fall survey to a national sample of 235 lower limb prosthesis users in a cross-sectional preliminary validation study, found survey questions to function as intended. Revisions to the survey were made at each stage of development based on analysis of participant feedback and data. The structured, 37-question lower limb prosthesis user fall event survey developed in this study offers clinicians and researchers the means to document, monitor, and compare fall details that are meaningful and relevant to lower limb prosthesis users in a standardized and consistent manner. Data that can be collected with the developed survey are essential to establishing specific goals for fall prevention initiatives in lower limb prosthesis users.

well as the Office of the Assistant Secretary of
Defense for Health Affairs, through the Orthotics
and Prosthetics Outcomes Research Program
(http://cdmrp.army.mil/funding/oporp) under
Award No. W81XWH-17-1-0551 (BH). Disclaimer:
The funders had no role in study design, data
collection and analysis, decision to publish, or
preparation of the manuscript. The opinions,
interpretations, conclusions and recommendations
are those of the author and are not necessarily
endorsed by Otto Bock Healthcare LP, Össur R&D,
or the Department of Defense.

**Competing interests:** The authors have declared
that no competing interests exist.

## 1.0 Introduction

Falls in lower limb prosthesis (LLP) users remain as common and consequential today as they
were over 20 years ago [1–8]. Between 1996 and 2001, 50 to 58% of LLP users reported
experiencing one or more falls in the previous year [1, 2, 6], while 21–29% of LLP users reported
some form of injurious fall over the same time period [2, 6]. More than 20 years later, 48 to 60%
of LLP users similarly report one or more falls a year [3–5, 9]. Likewise, estimates of the propor-
tion of LLP users experiencing one or more injurious falls a year remains between 18 and 26%
[7, 8]. Attempts to reduce the frequency of falls and their health-related consequences in LLP
users over the past 20 years have generally focused on: i) developing and validating clinical tests
to assess fall risk [4, 5, 10–12], ii) designing and testing prosthetic components to improve
patient safety [13–16], iii) quantifying biomechanical balance responses to identify deficits in
key balance recovery strategies [17–23], and iv) identifying modifiable and non-modifiable risk
factors [1, 2, 7, 8, 24, 25]. Given that issues related to falls continue to plague LLP users, alterna-
tive approaches to reduce falls in LLP users may be needed.

A key challenge to mitigating falls in LLP users may be illustrated by the public health
model [26], a systematic four-stage process that has been used to address falls in older adults
[27]. The first stage of the model is a comprehensive description of the problem (i.e., character-
izing the circumstances and consequences of fall events) [27]. Knowledge of how, where, and
when falls occur is intended to direct research in the model's later three stages; the identifica-
tion of risk factors, the design and evaluation of interventions, and the translation as well as
implementation of those interventions found to be effective [28]. The public health model
indicates that a thorough description of fall events is a critical first step, and without such
information there is little knowledge to inform the design, testing, and implementation of
effective fall prevention initiatives [3, 29].

Despite the importance of this foundational epidemiological data, little is known about the
circumstances and consequences of fall events in LLP users [2, 3, 6, 8, 29]. This gap can be attrib-
uted to the lack of a structured and clinically-meaningful fall event survey with which to solicit
details of LLP users' fall events [30]. While fall surveys have been developed for older adults [31–
42], they may not include details of falls experienced by LLP users. Prior work has illustrated
that LLP users' mobility experiences are unique [43], and their fall experiences are distinct [44].
In the absence of a structured survey to gather such information, past efforts to characterize fall
events in LLP users have been limited to data collected using ad hoc questionnaires [2, 6, 8, 16]
or unstructured interviews [3, 29]. Subsequently, the resulting fall circumstance and conse-
quence data lack the detail and consistency needed to advance fall prevention research and care
in this clinical population. A fall survey developed to be meaningful, clear, and applicable to LLP
users would greatly improve the quality and consistency of fall data collected in both research
and clinical care. An improved ability to document and therefore understand how, where, and
when LLP users fall would also help to direct fall prevention interventions to the most prevalent
and consequential types of falls [45–47], prioritize research needs related to fall risk assessment
[48], and generate evidence to develop and revise reimbursement policies. The objective of this
project was therefore to develop a structured, population-specific, fall survey capable of compre-
hensively characterizing the circumstances and consequences of fall events in LLP users.

## 2.0 Materials and methods

### 2.1 Overview

Best-practices in survey development and evaluation were used to design and test the lower
limb prosthesis (LLP) user fall event survey [43, 49–51]. All qualitative methods, and the

reporting of their results, adhered to published guidelines and standards [52, 53]. First, focus groups were conducted to identify scenarios and terminology central to the fall experiences of LLP users [44]. These data were used to develop meaningful fall event definitions, construct and revise a conceptual fall framework, create potential fall survey questions and response options that were meaningful, clear, and applicable to LLP users [50, 54–56]. Next, cognitive interviews were conducted to evaluate the clarity, comprehension, and applicability of the proposed fall event definitions, survey questions, and response options. Finally, in a preliminary validation study, a draft of the fall event survey was administered to a national sample of LLP users to test whether the survey functioned as intended (e.g., does the survey yield falls data consistent with expectations, and that available in the literature). Revisions to the survey were made at each stage of the study based on participant feedback, data collected with the fall event survey, and consensus among study investigators.

## 2.2 Participant recruitment and sampling

Focus group and cognitive interview study participants were recruited from across the United States via research registries, and print, email and Internet postings. Inclusion criteria included: i) lower limb amputation at or between the ankle and hip, ii) age greater than 18 years, iii) self-reported history of one or more falls in the past year, iv) use of a prosthesis, v) reported ability to speak and read English, and vi) agreement to have the discussions recorded and transcribed for subsequent analysis. Candidates were excluded if they could not complete a preliminary intake questionnaire or participate in a group discussion. Participants were purposively sampled [57] so that a range of perspectives might be solicited, thereby deepening our understanding of LLP users' fall experiences [58, 59]. Across all focus groups and for each set of questions tested in cognitive interviews, study investigators sought participation from at least two participants who were: i) transfemoral LLP users, ii) bilateral LLP users, iii) female, iv) greater than 50 years old, v) less than 1-year post-amputation, vi) of dysvascular amputation etiology, and vii) a Veteran or Service member.

Participants in the preliminary validation study were recruited from across the United States via a research registry and study flyers posted by professional and clinical partners. Quota sampling was used so that participants with a range of mobility levels could be surveyed. Specifically, participants were screened based on their Prosthetic Limb Users—Mobility (PLUS-M) T-score, with the goal of ensuring representation across all levels of mobility. Inclusion criteria for completing the fall survey included: i) lower limb amputation at or between the ankle and hip (in one or both legs), ii) age 18 years or older, iii) current use of a prosthesis, iv) use of a prosthesis for 6 months or more, and v) reported ability to speak and read English. No additional exclusion criteria were applied.

## 2.3 Data collection and analysis

**2.3.1 Focus group study.** Focus groups were convened to understand falls from the perspective of LLP users. Details outlining procedures for these focus groups were previously reported [44]. Briefly, focus groups were conducted via video conferencing to accommodate participants from across the United States. Focus group size was limited to 8 individuals in order to encourage input and discussion among all participants [60]. Seven open-ended guiding questions [44], conceived before conducting the focus groups and modeled after published guidelines [61], were used to promote discussion of shared experiences and vocabulary pertinent to fall events among LLP users [62]. To limit strong personalities from disproportionately driving the discussion, all participants were encouraged to voice their experiences [54, 63, 64].

Focus group discussions were transcribed in real-time to facilitate data analysis [65]. Focus groups were conducted until no new themes or concepts emerged [56, 66].

Six themes, found to characterize the fall experiences of LLP users, were used to construct a conceptual framework of falls in LLP users [44]. Formulation of these themes is described in detail elsewhere [44]. Briefly, an iterative and inductive process was used to review and organize focus group themes into higher level domains and identify potential relationships between each of the resulting domains [56]. In accordance with established standards [67], the ensuing fall framework was used to define content areas that would be measured by the survey. The framework also guided a review of existing fall surveys and related literature to help develop candidate items for the LLP user fall event survey. Gaps identified in the literature were addressed through the development of new questions and/or response options, subject to additional testing and revision following the cognitive interview and validation studies. Terminology used by LLP users in the focus groups to describe fall events (i.e., falls and near-falls) was also identified and used to create relevant and meaningful definitions that would be accessible to LLP users. Initial definitions were subject to further testing and revision during cognitive interviews.

**2.3.2 Cognitive interview study.** Cognitive interviews were conducted via telephone to accommodate local and national participants. Consistent with the development of other LLP user-specific surveys [51], retrospective verbal probing was used to evaluate survey instructions (including event definitions), questions, and response options [50, 51, 68, 69]. In this approach, each participant first completed an electronic copy of the initial fall event survey. Immediately after, verbal probing was used to solicit information about the thought processes used by each participant to answer select survey questions [70]. Five interview guides consisting of scripted open-ended questions (i.e., probes) were used to assess candidate survey questions and their response options for clarity (i.e., was the intended purpose of the question clear), comprehension (i.e., was the question understood similarly across participants), and applicability (i.e., could the question be answered using the given response options) [70] (S1 Appendix). Each interview guide included the same probes about survey instructions, fall event definitions, and fall history. Interview guides were limited to probing 4 to 6 of the survey questions to help participants remember how they arrived at their response, limit interview time, and ensure that each question was reviewed by at least 5 participants [62]. The 4–6 survey questions differed in each of the interview guides. The order of the questions within each interview was randomized. Areas of the survey from which questions and probes were developed were mixed across interview groups (e.g., questions about fall consequences were distributed across cognitive interview groups rather than presented in just one group). Interviews were audio-recorded and combined with field notes for subsequent analysis.

Following cognitive interviews, summaries of respondents' feedback were collated. Feedback that suggested fall event definitions were unclear or non-distinct was used to inform revisions to the definitions. Feedback that suggested the interpretation of survey questions and/or their response options differed among participants, varied from what study investigators intended, or found response options to be mutually exclusive or insufficient to answer a survey question was also used to inform revisions to the survey. Finally, feedback on aspects of balance and falls that respondents considered important, but were not included in the survey, was used to make additions to the survey. Revisions to survey content that included the addition, subtraction, substitution, or re-arrangement of a word or phrase that did not change the meaning of a question, or served to simplify the meaning of the question, were considered minor. Revisions in excess of these minor edits were considered substantial, and the relevant instructions, question(s) and/or response options(s) were subject to re-

assessment via additional cognitive interviews. Questions and response options that required no or only minor revisions, as well as those that were successfully revised and confirmed to function as intended, were included in the fall event survey that was administered in a preliminary validation study.

**2.3.3 Preliminary validation study.** A preliminary version of the fall event survey was then administered in a cross-sectional study to a national sample of LLP users. Study investigators sought to determine if survey questions operated as intended (i.e., data collected were consistent with expected patterns and existing literature), provided adequate coverage of fall-related circumstances and consequences experienced by LLP users, and included questions and response options that LLP users could recall. LLP users found to meet inclusion and exclusion criteria were sent a personal link to a secure REDCap application, where they could complete the survey online. Reminders to complete the survey were sent to potential participants up to four times. Participants were instructed to complete the fall survey based on their "most significant fall event" in the past year. This fall event was chosen with the intent of maximizing the confidence and accuracy with which fall circumstances and consequences were recalled by study participants [71]. "The most significant fall event" was also expected to increase the probability that study participants would need to endorse response options within the consequences section of the survey, serving therefore as a better "stress test" for the instrument than ones' "most recent fall".

To determine if survey questions operated as intended, Chi-square and McNemar tests were run to test for expected associations between fall circumstances and consequences (e.g., forward fall and impact with hands or knees), as well as expected patterns in survey responses (e.g., "*problem with prosthesis*" and "*wearing prosthesis*" were endorsed together). For statistical tests, α was set to .05. Statistical analyses were conducted using SPSS Statistics 28 software (IBM, Chicago, IL). The frequency of endorsement and content of participants' responses to the open-ended "*Other*" response option included with each question in the preliminary validation study survey was analyzed to assess and enhance the range of fall circumstances and consequences included in the fall event survey. When previously overlooked response options were suggested by multiple participants, and were not addressed in subsequent questions, they were added to the final survey. Finally, to determine whether the level of detail queried by the fall event survey could be recalled by LLP users, study investigators calculated: i) the frequency with which participants in the preliminary validation study endorsed the "*do not remember*" response option for each question, and ii) the percentage of participants who responded to the question "*how much confidence do you have in the details you provided*" by selecting *no confidence, low confidence, moderate confidence, high confidence, and complete confidence.*

For each study, demographic, health, and amputation characteristics were collected, along with Activities-specific Balance Confidence (ABC) scores [72] and Prosthetic Limb Users Survey—Mobility (PLUS-M) T-scores [73] to characterize participants' balance confidence and mobility, respectively. Measures of central tendency and dispersion, or frequency and proportion, were calculated to describe the demographic, health, amputation, balance, and mobility-related characteristics of the study samples.

The readability of final survey, and its constituent parts (i.e., instructions, definitions, questions and response options), was measured with the Flesh-Kincaid reading grade level [74]. Study protocols were reviewed and approved by institutional review boards at the University of Illinois at Chicago and the University of Washington. All individuals provided consent prior to participation.

## 3.0 Results

### 3.1 Focus group study: Terminology and the lived experience of lower limb prosthesis users

**3.1.1 Definition of fall and near-fall events.** Review and thematic analysis of focus group discussions guided by the question "*how would you describe a fall*?" revealed that participants described two types of similar events, falls and near falls. Focus group participants (n = 25, Table 1) [44] described how these events had a common element (i.e., both required a loss of balance), but ended with different outcomes (i.e., a fall ended with them being on the floor or ground, while a near-fall ended with them remaining on their feet).

Based on these descriptions, we created a model of these fall events with three distinct elements: i) the precursor, ii) the point of departure, and iii) the outcome (Fig 1). Across focus

**Table 1. Focus group (FG) study participant characteristics (n = 25).**

|  | FG 1 | FG 2 | FG 3 | FG 4 | FG 5 | Overall |
|---|---|---|---|---|---|---|
|  | n = 6 | n = 7 | n = 4 | n = 5 | n = 3 | n = 25 |
|  | n | n | n | n | n | n (%) |
| **Gender** |  |  |  |  |  |  |
| Male | 4 | 5 | 2 | 2 | 3 | 16 (64) |
| Female | 2 | 2 | 2 | 3 | 0 | 9 (36) |
| **Amputation level** |  |  |  |  |  |  |
| Bilateral (TT and TF) | 0 | 2 | 0 | 1 | 1 | 4 (16) |
| Transfemoral | 3 | 2 | 1 | 3 | 0 | 9 (36) |
| Transtibial | 2 | 5 | 3 | 1 | 3 | 14 (56) |
| **Amputation etiology** |  |  |  |  |  |  |
| Trauma | 2 | 3 | 2 | 1 | 4 | 12 (48) |
| Dysvascular | 2 | 1 | 0 | 2 | 1 | 6 (24) |
| Infection | 1 | 1 | 1 | 1 | 0 | 4 (16) |
| Cancer | 0 | 0 | 1 | 1 | 0 | 2 (8) |
| Other | 0 | 0 | 1 | 0 | 0 | 1 (4) |
| **Highest level of education** |  |  |  |  |  |  |
| Some college | 1 | 2 | 2 | 2 | 2 | 9 (36) |
| College degree | 2 | 4 | 1 | 0 | 1 | 8 (32) |
| Advanced degree | 3 | 1 | 1 | 3 | 0 | 8 (32) |
| **Other characteristics** |  |  |  |  |  |  |
| > 50 years old | 6 | 6 | 1 | 4 | 3 | 20 (80) |
| <1 yr prosthetic experience | 0 | 0 | 0 | 0 | 1 | 1 (4) |
| Military veteran | 1 | 2 | 1 | 1 | 1 | 6 (24) |
| $\geq$ 1 fall in past year | 5 | 7 | 3 | 4 | 3 | 22 (88) |
|  | Median (Median Absolute Deviation) | | | | | |
| Age (years) | 63.5 (6.5) | 64.0 (4.0) | 44.5 (7.0) | 66.0 (6.0) | 59.0 (5.0) | 59.0 (9.0) |
| Time since amputation (years) | 15.0 (7.0) | 18.0 (10.0) | 8.5 (3.0) | 44.0 (7.0) | 5.0 (4.0) | 17.0 (10.0) |
| Hours wearing prosthesis/day | 15.5 (0.5) | 15 (1.0) | 15 (1.5) | 16 (1.0) | 10 (1.0) | 15 (1.0) |
| Hours walking with prosthesis/day | 2.5 (0.5) | 4.0 (2.0) | 9.0 (3.0) | 8.0 (4.0) | 8.0 (0.0) | 6.0 (3.0) |
| PLUS-M (T-score) | 50.6 (3.2) | 51.2 (3.5) | 49.9 (3.5) | 53.6 (7.8) | 49.1 (9.3) | 51.2 (4.1) |
| ABC (0–4) | 2.81 (0.47) | 3.06 (0.32) | 2.50 (0.28) | 2.81 (0.56) | 2.19 (0.50) | 2.81 (0.57) |
| Number of falls in past year | 2.0 (1.0) | 2.0 (0.0) | 2.0 (1.0) | 3.0 (2.0) | 2.0 (2.0) | 2.0 (1.0) |

ABC: Activities-specific Balance Confidence scale; FG: Focus Group; hrs: hours; PLUS-M: Prosthesis Limb Users Survey—Mobility; TF: Transfemoral; TT: Transtibial

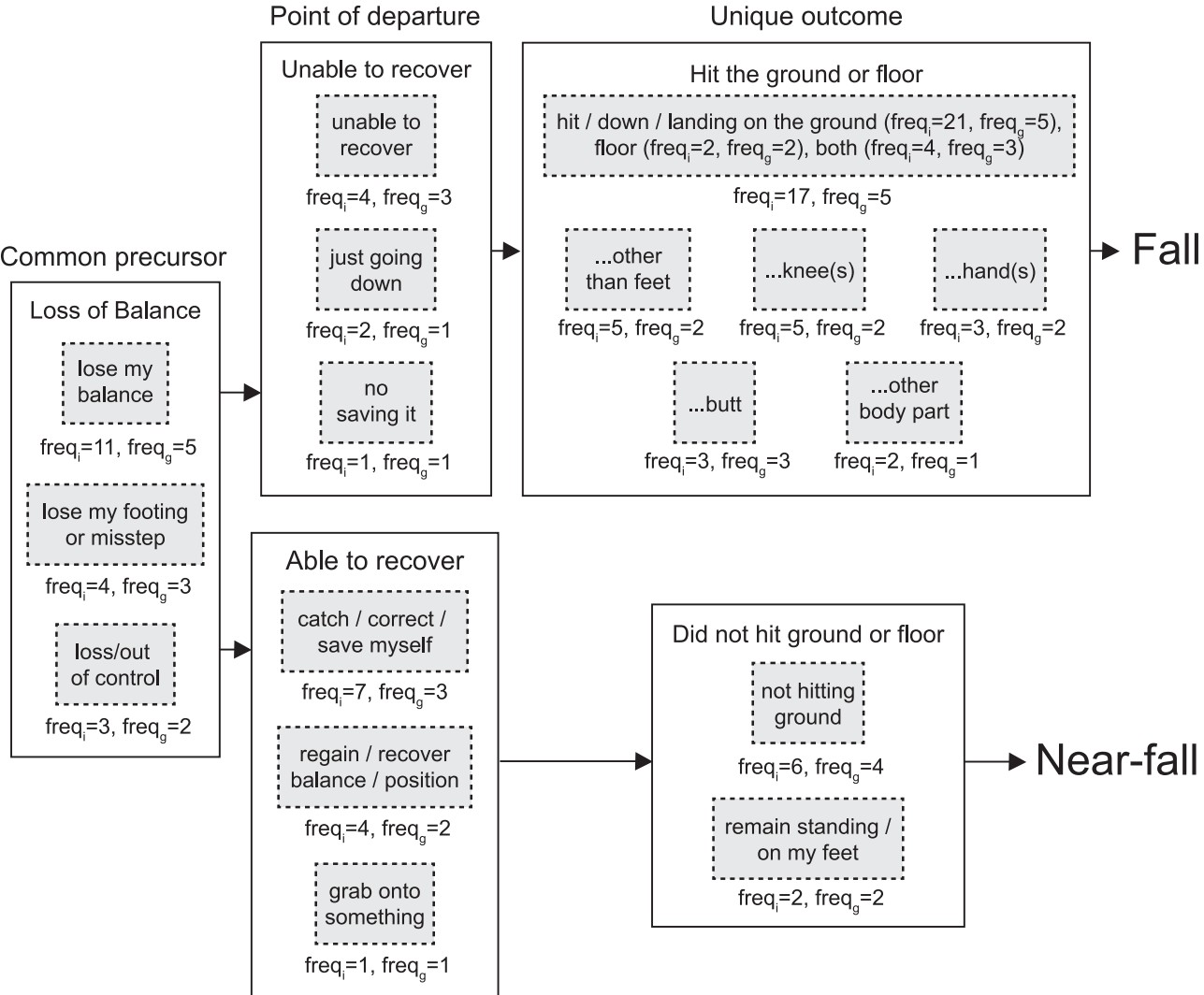

**Fig 1. Formulation of the fall event definitions.** Definitions for two overlapping yet unique fall events were proposed, tested, revised, and ultimately understood by lower limb prosthesis users. The focus group excerpts and the frequency with which they were used by the 25 individual lower limb prosthesis users (i.e., *freq_i*) regardless of focus group, as well as across the five focus groups (i.e., *freq_g*) are presented. Common to both falls and near-falls was a loss of balance. The ability or inability to recover marked the point of divergence between the two events, each ending in a unique outcome (i.e., hitting the ground / floor or not).

groups, participants described how a loss of balance was a common feature of falls and near-falls (i.e., a common precursor).

"*A fall is any time that I either lose my balance or lose my footing.*"

(gender: Male, age: 54 years old, level of amputation: bilateral transtibial (TT), time since amputation: 5 years since amputation)

"*I lose my balance and then have to grab on to something to stop me from hitting the ground.*"

(Female, 59 years old, TT, 8 years since amputation)

The ability to recover one's balance appeared to be the point at which falls and near-falls diverged for focus group participants. Participants consistently described a fall as an event where they were unable to recover from the loss of balance.

"*A fall is when there's no saving it.*"

(Male, 64 years old, bilateral TT, 43 years since amputation)

"*Once I start going down and I can't recover, to me that's a fall.*"

(Female, 57 years old, TT, 4 years since amputation)

In contrast, participants indicated that near-falls were those events where they were able to catch themselves and recover their balance.

"*If I catch myself, I do not consider that a fall*"

(Male, 81 years old, TF, 10 years since amputation)

"*I was thinking that the sensation you are losing your balance and could fall, but were able to recover*"

(Female, 57 years old, TT, 4 years since amputation)

Falls and near-falls were also described by focus group participants as having unique outcomes. A fall included contact with the floor or ground by parts of the body other than the feet. Some participants described specific examples of body parts (e.g., hands, knees) that made contact during a fall, while others were less specific.

"*Going all the way down to the ground is what I consider a fall*"

(Male, 59 years old, TT, 8 years since amputation)

"*When I have actually gone all the way to the ground, where I actually touched the knee or both knees to the ground is what I consider a fall*"

(Female, 52 years old, TF, 30 years since amputation)

Conversely, participants noted that with a near-fall, they did not contact the ground or floor.

"*I have lost my balance, but didn't actually, you know, end up on or against any other surface or floor*"

(Female, 43 years old, TT, 16 years since amputation)

"[A near-fall is when you] *Lost your balance and perhaps, you know, caught yourself before hitting the ground*"

(Male, 46 years old, TF, 50 years since amputation)

Notably, participants differentiated between the terms "floor" and "ground," which they felt related to indoor and outdoor surfaces, respectively[44]. These comments suggest that both terms should be included in any definition of a fall event.

"*The [phrase] "on the ground," indicates that it's an outdoor activity. People can fall indoors all the time. I did. And I didn't hit the ground. I hit my bedroom floor.*"

(Male, 76 years old, TT, 8 years since amputation)

Additionally, participants used colloquial terms such as slip, trip, and/or stumble interchangeably when describing fall events. These terms were also used when referring to both fall and near-fall events. Further, the perceived mechanics associated with terms like "slip" (e.g., direction your feet move relative to your body) differed between participants (e.g., "*feet would normally go back, maybe sideways*", "*forward*", "*Not really sure. Could it be any direction*?"). Study investigators therefore made a concerted effort to avoid colloquial terms such as slip, trip, and/or stumble in the fall event definitions as they may lead to inconsistencies in data collection and/or clinical assessments.

Based on these findings, the following initial definitions for falls and near-falls were proposed for the LLP user fall event survey: a *fall* is an accidental loss of balance where your body landed on the ground or floor, and a *near-fall* is an accidental loss of balance where you caught yourself or recovered your balance before your body landed on the ground or floor.

The clarity and comprehension of these initial definitions were evaluated based on cognitive interview participant feedback (n = 25 LLP users, Table 2). Revisions to each definition were made to resolve identified misinterpretations.

Cognitive interview participants almost universally agreed that the definitions were clear.

"*Yes, the definitions were clear.*"

(Female, 65 years old, TF, 10 years since amputation)

"*I understood each of the definitions. They were both straightforward.*"

(Female, 52 years old, TF, 32 years since amputation)

Participants were also able to describe how the two definitions differed, recognizing that a fall and near-fall are two distinct events, each with a unique outcome.

"*You were having me think about two different events. They end differently.*"

(Male, 69 years old, TF, 52 years since amputation)

"*I assume the distinction is between a temporary loss of balance that is recovered versus not being able to recover and landing on the floor.*"

(Male, 81 years old, TT, 12 years since amputation)

Despite general acceptance and a stated ability to distinguish between the two definitions, cognitive interview participants recommended minor modifications. Namely, words like "*accidental*" were deemed as implied, self-explanatory, or intuitive, and therefore unnecessary by more than half of cognitive interview participants.

"*Words like accidental are self-explanatory, maybe don't use it. I don't intend to fall.*"

(Female, 65 years old, bilateral: TT/TF, 11 years since amputation)

"*I'm not sure that accidental is necessary, seems implied.*"

**Table 2. Cognitive interview (CI) study participant characteristics (n = 25).**

| | CI 1 | CI 2 | CI 3 | CI 4 | CI 5 | Overall |
|---|---|---|---|---|---|---|
| | n = 5 | n = 5 | n = 5 | n = 5 | n = 5 | n = 25 |
| | n | n | n | n | n | n (%) |
| **Gender** | | | | | | |
| Male | 4 | 5 | 4 | 3 | 4 | 20 (80) |
| Female | 1 | 0 | 1 | 2 | 1 | 5 (20) |
| **Amputation level** | | | | | | |
| Bilateral (TT and TF) | 1 | 1 | 1 | 0 | 0 | 3 (12) |
| Transfemoral | 2 | 2 | 3 | 2 | 3 | 12 (48) |
| Transtibial | 3 | 3 | 3 | 3 | 2 | 14 (56) |
| **Amputation etiology** | | | | | | |
| Trauma | 2 | 3 | 2 | 1 | 4 | 12 (48) |
| Dysvascular | 2 | 1 | 0 | 2 | 1 | 6 (24) |
| Infection | 1 | 1 | 1 | 1 | 0 | 4 (16) |
| Cancer | 0 | 0 | 1 | 1 | 0 | 2 (8) |
| Other | 0 | 0 | 1 | 0 | 0 | 1 (4) |
| **Highest level of education** | | | | | | |
| Some college | 2 | 0 | 0 | 3 | 2 | 7 (28) |
| College degree | 2 | 3 | 2 | 2 | 2 | 11 (44) |
| Advanced degree | 1 | 2 | 3 | 0 | 1 | 7 (28) |
| **Other characteristics** | | | | | | |
| > 50 years old | 5 | 3 | 3 | 3 | 5 | 19 (76) |
| <1 yr prosthetic experience | 0 | 1 | 0 | 1 | 1 | 3 (12) |
| Military veteran | 0 | 1 | 3 | 0 | 2 | 6 (24) |
| ≥ 1 fall in past year | 3 | 4 | 5 | 2 | 4 | 18 (72) |
| | Median (Median Absolute Deviation) | | | | | |
| Age (years) | 68 (3.0) | 60 (9.0) | 66 (15) | 60 (5.0) | 64 (6.0) | 64 (5.5) |
| Number of co-morbidities | 2.0 (1.0) | 1.0 (0.0) | 1.0 (0.0) | 1.0 (1.0) | 0.0 (0.0) | 1.0 (1.0) |
| Time since amputation (years) | 10 (3.0) | 12 (3.0) | 25 (9.0) | 18 (0.0) | 19 (13) | 18 (8.5) |
| Hours wearing prosthesis per day | 10 (2.0) | 16 (1.0) | 15 (1.0) | 16 (1.0) | 14 (3.0) | 14 (2.0) |
| Hours walking with prosthesis/day | 3.0 (1.0) | 10 (5.0) | 7.5 (4.5) | 3.3 (1.8) | 5.0 (1.0) | 5.0 (3.0) |
| PLUS-M (T-score) | 56.3 (1.0) | 54.4 (5.2) | 47.7 (5.2) | 47.1 (4.9) | 55.3 (1.7) | 54.4 (3.4) |
| ABC (0–4) | 3.38 (0.13) | 3.44 (0.56) | 2.88 (0.38) | 2.40 (0.98) | 3.38 (0.63) | 3.16 (0.53) |
| Number of falls in past year | 1.0 (1.0) | 2.0 (2.0) | 1.0 (0.0) | 1.0 (1.0) | 2.0 (1.0) | 1.0 (1.0) |
| Number of near-falls in past year | 4.0 (1.0) | 4.0 (2.0) | 5.0 (1.0) | 2.0 (0.0) | 2.0 (2.0) | 3.5 (2.5) |

ABC: Activities-specific Balance Confidence scale; CI: Cognitive Interview; PLUS-M: Prosthesis Limb Users Survey—Mobility; TF: Transfemoral; TT: Transtibial

(Male, 60 years old, TT, 10 years since amputation)

Based on these recommendations, and a goal of keeping the definitions as brief and simple as possible, the word "accidental" was removed from the definitions. The fall and near-fall definitions were therefore revised to: a *fall* is a loss of balance where your body landed on the ground or floor, and a *near-fall* is loss of balance where you caught yourself or recovered your balance without landing on the ground or floor. The Flesch-Kincaid reading grade level of these final definitions was 3.9 for falls and 7.0 for near-falls. Both were reduced relative to the grade levels of the initial definitions (i.e., 6.3 and 10.9, respectively).

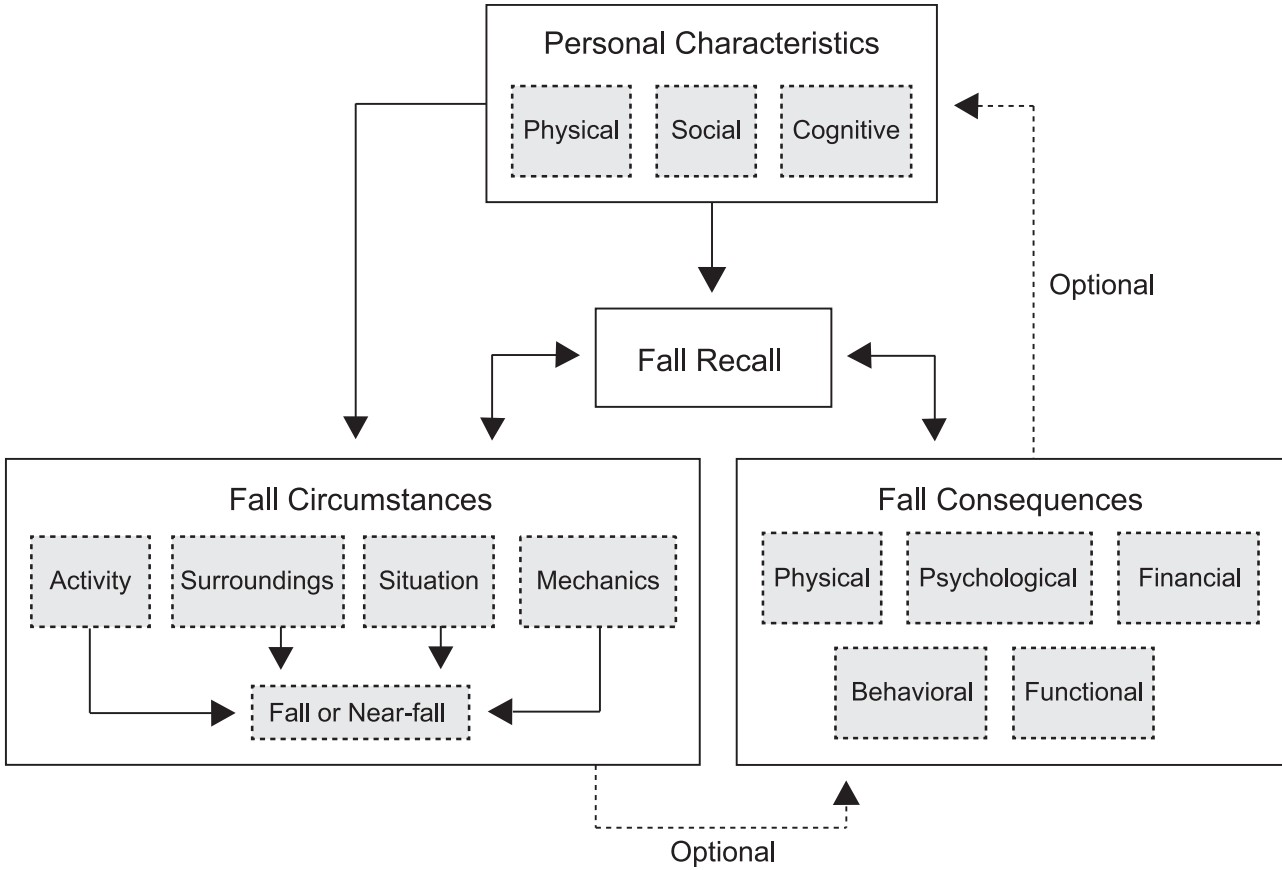

**Fig 2. Conceptual framework of fall events in lower limb prosthesis users.** The conceptual framework was used to identify, select, and test questions in the fall event survey that are relevant to LLP users.

**3.1.2 A conceptual framework of falls in LLP users.** A conceptual framework of falls in LLP users was developed based on the six themes identified from a previously-published qualitative analysis of focus group discussions with LLP users (Fig 2) [44]. The proposed framework consists of four primary components: personal characteristics, fall circumstances, fall consequences, and fall recall. The first component, personal characteristics, includes features of the individual that may affect the likelihood of a fall or near-fall. These include physical characteristics such as age, strength, and/or features of the amputation and prosthesis; cognitive factors such as attention and reasoning; social factors like living situation and education level, as well as emotional characteristics such as anxiety and depression. Personal characteristics are not assessed by the fall event survey, but rather via existing health questionnaires or scored self-report outcome measures. The second component of the framework, fall circumstances, refers to settings in which the fall event occurs or actions that take place prior to and during a fall event. These include the surroundings (e.g., immediate physical environment and social setting), activity (e.g., action being performed leading up to fall event), situation (e.g., conditions that may have disrupted the users balance or contributed to the fall) and fall mechanics (e.g., motions of the body during a fall event, and impact location). The third component of the framework, consequences, describes short- and long-term outcomes of the fall event. Consequences may be physical (e.g., injury), financial (e.g., medical costs, missed work), functional (e.g., change in how activities are performed), behavioral (e.g., change in the extent to which

activities are performed), and psychological (e.g., reduced balance confidence, embarrassment). As not all fall-related events will be associated with negative health outcomes, there may not always be consequences associated with an event. Finally, fall event recall, the fourth component, captures two temporal features within the framework, what LLP users remember about the event and for how long they remember the details of the event, up to and including remembering the event at all.

The components of the framework can act on and influence each other. For example, personal characteristics may influence the circumstances and consequences of a fall. If someone has osteoporosis, they may be more likely to break a bone in a fall, or if someone is a limited ambulator, they may be more likely to fall indoors. Not all falls will necessarily result in negative health outcomes, thus there may be no consequences to a fall. When they do occur, however, consequences may modify personal characteristics if they are of sufficient severity and/or persistence.

**3.1.3 Development of initial fall survey questions and response options.** Guided by the fall framework, a review of existing fall circumstance and consequence literature was performed to direct the development of survey questions and response options. Previous research regarding the circumstances of fall events were found to most often characterize the immediate physical surroundings (11 studies), followed by the fall mechanics (10 studies), activities being performed during a fall (7 studies), and situation (6 studies) immediately preceding or at the time of a fall [32, 35–42, 75, 76].

Details related to fall surroundings were found to have mainly focused on the location of the fall (e.g., indoors, outdoors) [35, 37–39, 41, 76]. Characteristics of the ground or floor (e.g., smooth, dry, icy, rough, uneven) were assessed less frequently, and often probed using specific examples (e.g., grass/leaves, cement/pavement) rather than general features (e.g., hard, uneven) [38, 39, 75].

Research on the activity performed at the time of a fall has largely concentrated on activity type (e.g., walking, turning, reaching, transfers) [32, 35, 36, 38, 41, 42, 75, 76]. The details of the activity (e.g., speed and direction) [76], or whether an assistive device (e.g., cane, walker) was being used at the time of the fall [32, 33, 75, 77] were rarely examined.

Attempts to characterize situation(s) contributing to a loss of balance were found to have included physiological [32, 37–39, 41] and mechanical disruptions [32, 37–39, 41, 75, 76] (e.g., dizzy, on medication, caught foot, slipped, misplaced step), as well as specific tasks that may distract or divert one's attention (e.g., texting, talking on phone) [32, 40, 75, 76]. Noted gaps in prior work included asking respondents about prosthesis-specific situations (e.g., prosthesis not on properly, prosthesis moved unexpectedly), as well as details about which leg was involved.

Interest in fall mechanics have generally focused on fall direction (e.g., forward, to the side) [32, 39, 40, 76], and to a lesser extent, impact location (e.g., hip, knees, hand) [35, 36, 39, 75]. Far less attention has been paid to the documentation of strategies intended to prevent/stop the fall and/or minimize risk of injury (e.g., modifying step placement, body rotation) [32].

With respect to consequences, immediate physical injury (e.g., fracture, contusion, concussion) [32, 36–38, 42, 75, 76, 78–81], followed by whether treatment was sought and/or received [36–38, 40, 41, 75, 76, 80, 82, 83], have been the main focus. Fewer studies have sought to describe the functional and behavioral consequences (e.g., changes in *how* or *what* activities are performed, respectively) [37, 41, 76, 84], psychological responses to fall events (e.g., decreased balance confidence, embarrassment) [40, 84], or whether the prosthesis incurred any damage.

Questions and response options were developed to address each of the content areas defined by the fall framework. Underserved areas identified in the literature, including but not

limited to surface characteristics and terrain grade; the speed and direction of activity; whether an assistive device was being used; the fit, function, or role of the prosthesis; which leg was involved; impact location(s); strategies to recover balance and/or minimize the risk of injury; as well as functional and emotional consequences of fall events were purposively targeted during question development.

The initial LLP user fall event survey produced after focus groups with LLP users consisted of 24 closed-ended questions, assessing the activity, surroundings, situation, mechanics, and consequences of fall-related events.

## 3.2 Cognitive interview study: Quality, clarity, and comprehension of survey content

Instructions, questions, and response options included in the initial fall event survey were revised based on feedback from cognitive interview participants (n = 25, Table 2).

**3.2.1 Instructions.** All 25 cognitive interview participants reported having read the instructions prior to answering the survey questions. Participants described the instructions as "*straightforward*", "*clear*", and "*[not] confusing.*" No suggestions for additional improvements were offered.

**3.2.2 Response options combined dissimilar alternatives.** Some response options were described by cognitive interview participants as confusing because they combined two distinct alternatives. For example, participants reported uncertainty when selecting the response option "*You were avoiding an obstacle or object*" because they indicated that obstacles were generally smaller than objects, and as such, different strategies would be used to avoid them (e.g., stepping over an obstacle versus moving around an object). Participants also reported difficulty selecting "*going up a ramp, hill, or incline*" because a hill would be outdoors, larger, and steeper, while a ramp would be manmade, indoors, shorter, and have only a moderate incline. Study investigators chose to resolve issues such as these by splitting the response option into two (e.g., going up a hill, going up a ramp).

**3.2.3 Response options were unclear.** Cognitive interview participants also identified issues of clarity, where their perceptions of responses options differed from what the study investigators intended. For example, the term "*sloped*" was interpreted as synonymous with ramp, hill, or uneven terrain, when it was intended to refer to a surface that slanted to the side, placing a LLP user's feet at unequal heights. To avoid confusion, study investigators removed this response option from the question "*what was the ground or floor like when you fell or nearly fell?*". Participants' view of several response options associated with the question "*What part(s) of your body hit the ground, floor, or other surface when you fell or nearly fell?*" also differed from what study investigators intended. For example, response options including "*lower arm*" and "*calf*" were interpreted to be any part of the arm below the shoulder, and the calf muscle rather than any portion of the leg between the knee and ankle, respectively. Study investigators clarified these response options by including explicit start and end points for each body part (e.g., *Arm—between the elbow and wrist)*.

**3.2.4 Response options were interpreted differently.** Opinions of response option meanings also occasionally differed among interview participants. For example, when asked about "*prosthesis moved unexpectedly,*" some respondents indicated that they would endorse this response option if their prosthesis became loose or they lost suspension due to a poorly fitting socket. In contrast, other participants said this response meant that their prosthetic knee buckled and "dropped them" to the ground. Consequently, responses could be influenced both by experiences related to socket fit and unexpected behavior of a prosthetic component at the time of the fall-event. To resolve any discrepancies between respondents, the phrasing of the

response option was revised to "*your prosthesis did not respond as intended*", and socket fit and suspension-specific response options were added (e.g., "*your socket was loose*").

**3.2.5 Response options were insufficient.** Cognitive interview participants described several instances where the available response options did not provide a sufficient level of detail, either because included responses were too broad or options were missing. For example, participants reported that possible answers to the question "*What part(s) of your body hit the ground, floor, surface or object when you fell or nearly fell*?" were missing. These included, face, prosthesis, and residual limb. Similarly, feedback indicated that response options to the question, "*Because of this fall or near-fall have you done or experienced any of the following*?" were limited. Specifically, they suggested that psychological consequences, such as "frustration," "anger," "depression" be included among the response options. For the same question, response options denoting changes in perceptions of balance confidence or fear of falling were viewed as overly narrow, failing to consider changes in one's confidence in their prosthesis. At the request of cognitive interview participants, and using their preferred terminology, study investigators addressed this gap by expanding the response options to include these suggestions. To accommodate the greater number of response options, study investigators created separate questions for emotional consequences (e.g., depression, frustration), behavioral consequences (change, avoid, or stop certain activities), and other fall-related consequences (e.g., decreased balance confidence, more afraid of falling). Feedback from participants also revealed that several response options for the question "*Did you seek medical treatment because of your fall or near-fall*?" were missing. Participants suggested that "physical therapy," "massage therapy," and "repair or replacement of their prosthesis" could be treatments sought after a fall event. These options were all added. Finally, participants noted that the question related to activity at the time of a fall should include showering or bathing, as they described feeling particularly vulnerable to falls when the bathroom surface was wet, and they needed to remove their prosthesis. Showering or bathing was therefore added as a response option to the question "*what were you doing when you fell or nearly-fell*?".

**3.2.6 Overall survey content.** When asked "*What other aspects of balance and falls do you think should be included in the survey*?*",* and "*What kinds of things are important for us to ask about if we want to learn more about falls*?" participants generally indicated that the fall survey was comprehensive (e.g., "*nothing else I can think of*", "*seemed to cover all bases and be fairly extensive*"). Other participants provided specific suggestions better addressed through data collected in conjunction with, but not as part of, the LLP user fall survey (e.g., "*issues of age*", "*footwear*", "*type of prosthesis*", "*time since amputation*"). Lastly, a few respondents emphasized the psychological consequences of falls (e.g., "*fear of falling*", "*ask more about confidence or tentativeness*", "*inquire about any emotional or psychological changes*"). These recommendations were addressed through the expansion of response options associated with questions pertaining to fall-related consequences described above. After all revisions based on cognitive interviews, the fall event survey consisted of 30 questions. This version of the survey was then administered to a national sample of LLP users in a preliminary validation study to further evaluate its content and function.

Based on cognitive interview participant feedback, instructions were deemed acceptable in their current form, twelve questions and their associated response options were unchanged, while one question was removed, and two were added based on feedback from cognitive interview participants. New response options were added to ten questions, while existing response options were deleted for three questions, revised for seven questions, and moved for one question. Finally, two questions and their response options were split up into six more focused questions.

### 3.3 Preliminary validation study: Cross-sectional assessment of fall event survey function

**3.3.1 Participant characteristics.**    235 LLP users participated in the study. Participant demographic, health, amputation, mobility, and balance characteristics are reported in Table 3. 212 participants (90.2%) recalled one or more falls and/or near-falls in the past year and completed the fall event survey. Median time to complete the fall event survey was 11 minutes, with an interquartile range of 6 minutes (Q1: 8 minutes, Q3: 14 minutes). Among them, 168 participants (71.5%) recalled one or more falls in the past year, 191 or 81.3% recalled one or more near-falls in the past year, and 23 or 9.8% recalled no falls or near-falls in the past year. Among the 212 participants who recalled a fall and/or near-fall in the past year, 158 reported that a fall was their most significant event in the past year, while 54 reported that their most significant event was a near-fall (S1 Dataset).

**3.3.2 Fall survey operated as intended.**    Expected associations between fall circumstances and/or consequences across survey questions were observed (e.g., forward fall and impact with hands or knees). Chi-square and McNemar tests revealed that anticipated associations between fall direction and impact location, injury level and treatment sought or received, as well as fall circumstances and the use of a prosthesis or assistive device were statistically significant ($X^2 \geq$ 5.32, p $\leq$ .021) (Table 4, S1 Dataset). Small to medium effect sizes (phi ($\phi$) = .158 to .444) were observed for these statistically significant associations (Table 4).

**3.3.3 Survey participants confidently recalled details of the most significant fall-related event.**    The "*do not remember*" response option was selected infrequently by study participants. On more than two thirds of survey questions, 5% or less of the participants who reported a fall as their most significant event in the past year (i.e., fall participants) endorsed "*do not remember*". Across survey questions, the percentage of fall participants who endorsed "*do not remember*" ranged from 0 to 6%. "Do not remember" responses were endorsed more frequently for activity-related questions (range: 3% to 6%), and less frequently for consequence-related questions (0% to 2%). Among the surrounding, situation, and fall-related mechanics questions, "*do not remember*" was endorsed by <1% to 2%, 0% to 5%, and 4% to 5% of fall participants, respectively (S1 Dataset).

Participants who reported a near-fall as their most significant event in the past year (i.e., near-fall participants) selected "*do not remember*" with a slightly higher frequency than fall participants. On more than half of the survey questions, 5% or fewer of the near-fall participants endorsed "*do not remember*". Endorsement of "*do not remember*" exceeded 10% of respondents on just two questions. Across survey questions, the percentage of near-fall participants who endorsed "do not remember" ranged from 0% to 15%. Similar to fall participants, near-fall participants endorsed "*do not remember*" more frequently for activity-related questions (range: 4% to 15%), and less frequently for consequence-related questions (0%). Among the surroundings, situation, and fall-related mechanics questions, "*do not remember*" was endorsed by between 6% to 9%, 2% to 6%, and 0% to 9% of near fall participants, respectively.

When asked at the end of the survey "*how much confidence do you have in the details you provided about the fall or near-fall*", most fall (82%) and near-fall (81%) participants reported either "*high confidence*" or "*complete confidence.*" Fewer fall (16%) and near-fall (17%) participants reported having "*moderate confidence*" in the details they reported (S1 Dataset).

**3.3.4 Survey content was expanded based on participants' "Other" responses.**    Analysis of the open-ended "*Other*" response fields revealed that additional response options were needed to fully characterize the fall circumstances and consequences experienced by LLP users. Among questions within each of the survey sections (i.e., surroundings, activity), an average of 2 to 16% of study participants endorsed the "*Other*" response option. Many of the

**Table 3. Preliminary validation study participant characteristics (n = 235).**

| | Most significant fall event in the past 12 months | | |
|---|---|---|---|
| | No fall event reported (n = 23) | Fall (n = 158) | Near-fall (n = 54) |
| | Number of participants (% of whole sample, n = 235) | | |
| **Gender** | | | |
| Male | 18 (8%) | 92 (39%) | 30 (13%) |
| Female | 5 (2%) | 66 (28%) | 24 (10%) |
| **Amputation level** | | | |
| Unilateral transtibial | 15 (6%) | 78 (34%) | 19 (8%) |
| Unilateral transfemoral | 4 (2%) | 48 (20%) | 19 (8%) |
| Bilateral (TT and TF) | 3 (1%) | 15 (6%) | 9 (4%) |
| Other (e.g., HD) | 1 (<1%) | 17 (7%) | 7 (3%) |
| **Amputation etiology** | | | |
| Dysvascular | 10 (4%) | 60 (26%) | 14 (6%) |
| Non-dysvascular | 13 (5%) | 98 (42%) | 40 (17%) |
| **Highest level of education** | | | |
| Some high school | 0 (0%) | 43 (18%) | 18 (8%) |
| High school graduate | 3 (1%) | 54 (23%) | 22 (9%) |
| Some college | 7 (3%) | 10 (4%) | 4 (2%) |
| College degree | 9 (4%) | 1 (<1%) | 0 (0%) |
| Advanced degree | 4 (2%) | 43 (18%) | 8 (3%) |
| **Employment status** | | | |
| Employed | 10 (4%) | 3 (1%) | 1 (<1%) |
| Retired | 6 (3%) | 17 (7%) | 5 (2%) |
| Unemployed | 1 (<1%) | 50 (21%) | 21 (9%) |
| Student | 0 (0%) | 46 (20%) | 17 (7%) |
| On disability | 6 (3%) | 42 (18%) | 10 (4%) |
| Homemaker | 0 (0%) | 3 (1%) | 1 (<1%) |
| **Daily use of assistive devices** | | | |
| One cane | 7 (3%) | 21 (9%) | 16 (7%) |
| Two cane | 1 (<1%) | 2 (<1%) | 1 (<1%) |
| Two crutches | 6 (2%) | 1 (<1%) | 5 (2%) |
| Walker | 5 (2%) | 9 (4%) | 12 (5%) |
| Wheelchair | 18 (8%) | 27 (11%) | 22 (9%) |
| **Fall events in past year** | | | |
| None | 7 (3%) | 107 (46%) | 16 (7%) |
| ≥ 1 fall | 0 (0%) | 158 (67%) | 19 (8%) |
| ≥ 1 near-fall | 0 (0%) | 141 (60%) | 54 (23%) |
| | Median (Median Absolute Deviation) | | |
| Age (years) | 57.3 (12.0) | 60.7 (7.61) | 61.4 (6.01) |
| Number of co-morbidities | 1.0 (1.0) | 1.0 (1.0) | 1.0 (1.0) |
| Number of daily medications | 4.5 (5.5) | 4.0 (3.0) | 2.0 (2.0) |
| Time since amputation (years) | 11.9 (3.91) | 12.7 (8.44) | 19.0 (9.24) |
| Hours wearing prosthesis wear/day | 15.0 (1.50) | 14.0 (2.00) | 14.0 (2.00) |
| Hours walking with prosthesis/day | 5.50 (2.00) | 3.00 (2.00) | 4.00 (2.00) |
| PLUS-M (T-score) | 57.0 (6.08) | 50.3 (7.50) | 52.8 (8.06) |
| ABC (0–4) | 3.60 (0.47) | 3.13 (0.41) | 3.63 (0.50) |
| Number of falls in past year | 0.0 (0.0) | 2.0 (1.0) | 2.0 (1.0) |

(*Continued*)

**Table 3.** (*Continued*)

| | Most significant fall event in the past 12 months | | |
|---|---|---|---|
| | No fall event reported (n = 23) | Fall (n = 158) | Near-fall (n = 54) |
| Number of near-falls in past year | 0.0 (0.0) | 3.0 (2.0) | 4.0 (2.0) |

ABC: Activities-specific Balance Confidence scale; CI: Cognitive Interview; PLUS-M: Prosthesis Limb Users Survey—Mobility; TF: Transfemoral; TT: Transtibial

"*Other*" responses provided by study participants were however addressed by questions later in the survey, and were therefore deemed not to require revision, but rather repositioning within the survey. For example, for the question, "*what were you doing when you fell or nearly fell*?", several participants reported "*using crutches*" or "*walking with a cane*". The use of an assistive device at the time of a fall or near-fall was originally queried much later in the survey. The questions were subsequently re-ordered, moving the assistive device question to a point earlier in the survey. Similarly, the question asking whether a prosthesis was worn at the time of a fall or near-fall, was moved to the start of the survey. In total, 11 additional response options were added to six questions (Table 5, S1 Dataset). For example, in response to the question, "*Did any of the following occur to you when you fell or nearly fell*?", participants wrote in "*stepping on an unseen object*", and "*did not notice a small object on the floor and I stepped on it*". Initial response options did not cover this situation, consequently, "*stepped on something*" was added as a response option to this question. With respect to fall mechanics, a host of participants described strategies to minimize the risk of injury that were not included in the original response options. These included, "*I tucked and rolled to avoid injury*", and "*Put my hands out to avoid landing on my residual limb*". "*Tucked and rolled*" and "*Used arms to brace yourself*" were therefore added as response options to the question, "*Did you do anything to minimize the risk of injury*?". For the final survey, the open-ended "other" response option accompanying each question was deleted and replaced with an open-ended question at the end of each section of the survey. For example, at the end of the activity section of the survey participants are asked, "*Is there anything else you would like to tell us about what you were doing when you fell or nearly fell*?". Similar questions were added to the end of the surroundings, situation, fall mechanics, and consequences sections.

**Table 4. Expected associations between response options endorsed by study participants reporting a significant fall-related event in the past 12 months (n = 212).**

| Response option comparisons | $X^2$ statistic | p-value | phi (ϕ) |
|---|---|---|---|
| Q19: "*Problem with prosthesis*"** and Q1: "*Wearing prosthesis*" | 6.69 | .015* | .444 |
| Q3: "*Walking, stepping, or running*" and Q2: "*Wheelchair or scooter*" | 9.67 | .002* | -.214 |
| Q4: "*Moving quickly*" and Q18: "*Hurried or rushed*" | 11.2 | .002 | .229 |
| Q19: "*AD broke or moved unexpectedly*" and Q2: "*Using AD*" | 5.32 | .021 | .158 |
| Q9: "*Unfamiliar*" and Q18: "*Distracted, not paying attention*" | 1.23 | .268 | .139 |
| Q17: "*Foot got caught*" and Q5: "*Stepping over something*" | 17.4 | .001 | .287 |
| Q21: "*Fall forward*" and Q25: "*Wrist, hand, elbow, or knee impact*" | 9.40 | .002 | .211 |
| Q21: "*Fall backward*" and Q25: "*Buttocks, hip, or head impact*" | 25.4 | .001 | .346 |
| Q28: "*Major injury*"*** and Q29: "*Treatment sought*" | 37.5 | .001 | .421 |
| Q28: "*Major injury*" *** and Q30: "*Treatment received*" | 40.8 | .001 | .438 |

AD: Assistive device (i.e., cane, crutch, or walker)

\* Expected cell count < 5, Fisher's Exact test conducted.

** Problem with prosthesis includes "prosthesis broke", "prosthesis did not respond as intended", and "prosthesis was not on properly"

*** Major injury includes "fracture or broken bone", "internal injury", or "concussion or head injury"

**Table 5. Response options added to the fall survey based on open-ended "Other" field responses provided by study participants during the small-scale administration.**

**Question 5. Were you doing any of the following when you fell or nearly fell?**

| Other open-ended response(s) | Response option(s) added/revised |
|---|---|
| "*Lifting a package into the car*" | Lifting or carrying something |
| "*Picking something up*" | |

**Question 17. Did any of the following occur to you when you fell or nearly fell?**

| Other open-ended response(s) | Response option(s) added/revised |
|---|---|
| "*Did not notice a small object on the floor and I stepped on it with my foot*" | Stepped on something |
| "*Didn't pay attention to small object on the ground*" | |
| "*I stepped on an unseen object on the ground*" | |
| "*Knee collapsed*" | Left / right leg gave out |
| "*My sound knee gave out*" | |
| "*My non-prosthetic knee just buckled*" | |

**Question 19. Did any of the following contribute to the fall or near-fall?**

| Other open-ended response(s) | Response option(s) added/revised |
|---|---|
| "*I walked out of my socket*" | Prosthesis came off |
| "*My prosthesis came off*" | |
| "*My leg fell off*" | |
| "*Handrail broke away*" | Something you were holding moved or gave way |
| "*Walker slipped out from under me*" | |

**Question 22. Did you do anything to catch yourself or prevent the fall or near-fall?**

| Other open-ended response(s) | Response option(s) added/revised |
|---|---|
| "*I moved and waved my arms around to try and catch my balance*" | Moved or waved arms around |

**Question 23. Did you do anything to minimize the risk of injury?**

| Other open-ended response(s) | Response option(s) added/revised |
|---|---|
| "*I tucked and rolled to avoid damage*" | Tuck and roll |
| "*I rotated to the side to impact my sound rather than prosthetic leg*" | Rotate to the right / left |
| "*Put my hands out to avoid landing on my residual limb*" | Used arms to brace yourself |
| "*Used my legs to ease down slowly*" | Eased yourself down |
| "*Slowed my descent with my legs*" | |

**Question 30. Did you receive medical treatment because of the fall or near-fall?**

| Other open-ended response(s) | Response option(s) added/revised |
|---|---|
| "*I am an RN, so I tended to my own wounds*" | Self-administered treatment |
| "*I administered my own first aid*" | |

Following all revisions, the final LLP user fall event survey consisted of 37 questions assessing the activity, surroundings, situation, mechanics and consequences of fall related events. The median Flesh-Kincaid reading grade level of the final LLP users fall event survey was 5.3, with a median absolute deviation of 1.6. Reading grade level of the survey instructions was 7.6. Situation and consequence-related questions had the highest median grade level (6.2), followed by fall mechanic questions (5.2), activity-related questions (3.6), and surrounding-related questions (1.0).

## 4.0 Discussion

The objective of this project was to develop a structured, population-specific, fall survey capable of comprehensively characterizing the circumstances and consequences of fall events in

LLP users. Using best practices in survey development, including focus groups and cognitive interviews to solicit input from target respondents, we developed a 37-question survey to characterize fall frequency, circumstances, and consequences. The proposed LLP users fall event survey (S2 Appendix) offers clinicians and researchers a means to consistently document and compare fall events among their participants and patients. The comprehensive data that can be collected with the fall survey are critical to establishing specific goals for fall prevention interventions in LLP users. Below, three questions that are central to the administration of the LLP user fall event survey are discussed: i) why should the survey be trusted? ii) what advantages does the survey have when compared to existing approaches? and iii) how might the survey be used?

## 4.1 Why should the LLP user fall event survey be trusted?

The methods used, stakeholders involved, results obtained, and revisions made during survey development and testing were intended to impart confidence in the ability of the LLP user fall event survey to obtain valid and meaningful data. Best-practices in survey development and evaluation were used to design and test the structured LLP user fall event survey [43, 49–51]. Focus groups conducted with a diverse sample of LLP users identified scenarios and terminology central to the fall experiences of LLP users [44]. Documentation of the lived experience guided the construction of a conceptual fall framework (Fig 2), an exercise that in combination with a thorough review of the relevant literature helped ensure survey content covered areas that are meaningful and relevant to the target population of LLP users. Previously overlooked content areas that were included in the current survey were details of the activity at the time of the fall, strategies to prevent the fall and/or minimize injury, psychological responses to a fall event, as well as LLP user-specific scenarios. The decision to distinguish between and define both falls and near-falls was also guided by focus group input from LLP users. Near-falls were described by LLP users to have the potential to be as consequential as falls. Near-falls were therefore understood to be meaningful events for LLP users that merit consideration equal to that of falls, yet rarely receive it [15, 16].

The feedback obtained from LLP users during cognitive interviews further demonstrates that survey instructions, questions, and response options were clear, applicable, and well understood by LLP users. Importantly, participants agreed that the fall event definitions were clear, and represented overlapping yet unique events. Revisions to survey content based on cognitive interview participant feedback should also increase confidence in the survey, as changes largely involved just adding additional response options to improve survey applicability or splitting existing response options to enhance survey clarity and comprehension. Improvements to the clarity and comprehension of the fall survey were achieved by splitting response options perceived by LLP users to consist of unique, non-overlapping answers. Adding to the clarity and comprehension of the fall survey was the deliberate effort by study investigators to use accessible language and phrasing throughout. Following all revisions based on participant feedback, the Flesch-Kincaid reading grade level for each section in the final survey was below the recommended level of 5.3 for written health materials [74, 85].

Results from the preliminary validation study, conducted with a large national sample of LLP users, should also impart a degree of confidence in the LLP user fall event survey. First, the relatively low frequency with which participants endorsed the "*Other*" response option for any given question served to confirm the breadth and depth of the fall survey (i.e., it provides sufficient coverage of relevant fall-related circumstances and consequences). Generally, when the "*Other*" response option was endorsed and respondents provided details of the event, these were often already included later in the survey.

Second, the low frequency with which survey respondents endorsed "*do not remember*" across fall survey questions (i.e., <5% on average), and the high level of confidence they expressed in their recollection of the fall event (i.e., >80% reported a high level of confidence) suggests that the questions and accompanying response options in the survey represent fall-related circumstances and consequences that LLP users can confidently recall. The confidence with which the circumstances and consequences were recalled in the current study may have been due in part to the nature of the fall event that was probed (i.e., "*the most significant fall event in the past 12 months*"). It is possible, if not likely, that less significant fall events are not as memorable, and LLP users would experience more difficulty in recalling the details surrounding those events.

Confidence in the survey as a tool for documenting fall events by LLP users was further bolstered when survey questions were found to yield falls data consistent with clinical expectations, and that available in the literature [2, 3, 29] (e.g., forward fall and impact with hands and/or knees). Had the preliminary validation study produced fall circumstance and/or consequence data that did not align with expectations, confidence in its administration would be questioned, and its efficacy to document fall events in LLP users in doubt. Overall, the methods used, results obtained, and revisions made indicate that survey definitions, questions, and response options are clear, well understood, comprehensive, and applicable to LLP users. These outcomes should inspire confidence among clinicians and researchers when administering the LLP user fall event survey.

## 4.2 What advantages does the LLP user fall event survey have when compared to existing approaches?

The structured and comprehensive design of the LLP user fall event survey offers several advantages for documenting fall events over conventional ad hoc fall questionnaires [2, 6, 8, 76] and unstructured interviews [3, 29]. Overall, the fall event survey provides a tool to collect fall event data comprehensively and consistently. Administration of the survey is therefore likely to yield comparable data, which can be aggregated and/or compared between fall studies, study sites, or clinics, as well as within or between individual LLP users for clinical decision making. The collection of quality data begins with clear and meaningful definitions of fall events. Without a clear definition, LLP users may interpret the meaning of a fall in different ways [86]. Data collected to characterize fall events and their consequences using vague definitions could subsequently be based on different events, limiting comparability and thus utility for understanding falls in LLP users. Historically, most but not all studies [1, 6, 87, 88] that have sought to document the frequency, circumstances and consequences of falls, or identify potential fall risk factors in LLP users, have included a fall definition. Variation in fall definitions across studies, the inclusion or omission of terminology the current study found problematic (e.g., "*unintentional / inadvertent*", "*ground or floor*", "*comes to rest on*", "*stumble*"), as well as the absence of a near-fall definition, suggests that there may be potential challenges in comparing results between studies. The LLP user fall survey addresses these challenges and increases the consistency with which falls data can be collected by including fall event definitions that are based on input and feedback from LLP users, avoid confusing colloquial terminology, and clearly differentiate falls from near-falls.

Further enhancing the consistency with which falls data can be collected with the fall event survey, and in contrast to most [2, 3, 6, 8, 29] but not all [76] previous efforts to characterize fall circumstances and consequences in LLP users, the LLP user fall event survey uses a set of closed- rather than open-ended questions. The use of fixed-response options ensures that all respondents select from the same set of possible response options for each question. As a

result, data is easily aggregated across respondents, enabling analysis and interpretation [57]. Closed-ended questions also limit vague or incomplete responses and facilitate easier comparisons between studies. Perceived or potential drawbacks associated with closed-ended questions (i.e., too narrow in scope) are offset by the comprehensive and detailed set of response options that were developed and tested based on the lived experience of LLP users. To address the potential for details not included in the survey, each section ends with a single open-ended question, where respondents are asked: "*Is there anything else you would like to tell us about (insert module topic, e.g., what you were doing) when you fell*?". As a result, the consistency offered by many fixed response options is balanced with the flexibility of a single "catch all" open-ended question capable of accommodating any unique or novel fall circumstances and/ or consequences.

In addition to enhancing consistency and comparability, the LLP user fall event survey generates a richness of data that exceeds what has previously been reported. The breadth and depth with which fall events among LLP users can be characterized with this survey is expected to yield actionable information for a wide range of stakeholders. The quantity and quality of the fall event data may produce evidence that informs new and/or serves to modify existing reimbursement and prescription policies for prosthetic components intended to reduce falls and fall-related injuries (e.g., prescription of microprocessor knees for limited community ambulators). Administration of this survey could also aid in establishing the most pressing research needs regarding fall assessment and prevention by shifting attention from all-cause falls, towards specific and consequential types of falls. The breadth and depth of the data that can be collected with the fall event survey is also likely to yield previously unreported fall details, which may generate novel hypotheses that can be tested. For example, the preliminary validation study showed that the most-commonly endorsed response to "*did you do anything to catch yourself or prevent the fall*", was "*reached out to grab someone or something*". Compared to the more heavily researched stepping response [17, 18, 20, 89–91], the reach and grasp strategy was endorsed by nearly a five-to-one margin. A complete accounting of fall events may therefore point to the need for alternative experimental approaches to study the biomechanics of falls in LLP users. Data collected with this survey may also allow manufacturers to develop and test prosthetic components designed to respond to the most prevalent and consequential types of falls. Clear and meaningful definitions, fixed-response options, as well as the comprehensiveness of fall data that can be collected with the LLP user fall event survey has the potential to make large, timely, and important contributions to limb loss science and clinical care.

Identifying existing and establishing additional benefits of the LLP user fall event survey is an on-going area of research. A direct comparison between data collected with the fall event survey and a conventional patient interview would serve to clarify the "value added" of using a standardized fall survey over less structured approaches. The translation and validation of the LLP user fall event survey into other languages and cultures would also add to its value in the prosthetic community by increasing access to the survey internationally and ensuring valid and meaningful comparisons of fall events across cultures and languages. Prospective administration of the fall event survey (i.e., administering the survey immediately after falls occur) would serve to better characterize the range of fall events experienced by LLP users, and expand upon knowledge about the "most significant" fall events we measured in the preliminary validation study. Finally, additional research to expand and revise survey content to reflect fall experiences of other clinical populations (e.g., stroke, multiple sclerosis) would uniquely position researchers to compare important details of fall events between clinical populations and highlight areas of overlap as well population-specific needs with a single instrument.

## 4.3 How might the LLP user fall event survey be used?

The fall event survey has potential applications across each of the four stages of the public health model [27]. Having resolved the need for a structured and standardized approach to comprehensively characterize fall events in LLP users in the current study, it is now possible to address gaps in stage one of the public health model as applied to falls by LLP users; "describe the problem" [27]. The fall event survey may also prove useful in stage two, interpreting factors that increase or decrease the risk of falls. Rather than regarding risk factors as generic, and related to any and all falls, risk factors may instead be associated with specific fall event outcomes (e.g., injurious versus non-injurious), and/or fall event circumstances (e.g., forward versus backward fall directions). The administration of the survey during research focused on causes and correlates of fall events in LLP users (i.e., stage 2 of the public health model) may therefore identify factors that serve to motivate the design of fall prevention initiatives, fall prediction measures, or prosthetic components that are specific to particular fall types. Finally, in stages 3 and 4, the fall event survey could help track fall event outcomes during comparative effectiveness trials. Here, the level of detail captured by the survey could be useful in helping determine whether fall prevention interventions (e.g., initiatives or devices) are broadly effective regardless of fall circumstances, or more effective at reducing a specific type of fall. Establishing the degree of specificity associated with an intervention may contribute to improved prescription guidelines and prevent an intervention that would otherwise be abandoned if found to be ineffective at reducing all-cause falls.

As wearable sensors become more prevalent, and interest in their use to detect [92, 93] as well as understand the movements of real-world falls becomes of increasing interest [86, 94–96], the LLP user fall event survey is uniquely suited to provide the context required to interpret the voluminous and noisy data wearable sensor generate. In fact, descriptions of the circumstances surrounding falls using voice recorders were critical to interpretation of otherwise noisy data collected with inertial measurement units during fall events in older adults [97]. Administering the fall event survey in concert with the deployment of wearable sensors may help explore key movements during real-world fall events in LLP users. In doing so, the same data may contribute to concurrent improvements in the sensitivity and specificity of algorithms intended to provide automated detection of fall events in LLP users from wearable sensor data [92, 93, 98].

The LLP user fall event survey can be used to collect retrospective or prospective fall event data during cross-sectional and longitudinal studies, respectively. When collecting *retrospective* fall data in a cross-sectional study, researchers should consider the length of the recall period and which fall event(s) to document, as both may influence fall recall decay. Given the level of detail within the survey, and our limited understanding of fall recall decay in LLP users [5], it may be prudent to focus on a single memorable fall event, particularly if a longer recall period is selected (e.g., one year). Concentrating on a single fall event is likely to maximize respondent recall and therefore data accuracy. Memorable fall events that may be of interest and advance our understanding of falls in LLP users include, but are not limited to, the most recent fall, the most injurious fall, or as was done in the current study, the fall event deemed most "significant' by study participants. When collecting *prospective* data, researchers often seek to minimize fall recall decay by limiting the time that elapses between when a fall occurs and when it is documented. Current recommendations suggest that prospective fall events be recorded daily (i.e., fall or no fall, details of the fall), and reported on a monthly basis [99]. Several of the survey questions that document the consequences of a fall event may however require additional time after a fall before they can be answered (i.e., questions 29–35). Depending on the nature and severity of the fall event, a three-to-seven-day delay may need to be built

into the timeline between the fall event and its documentation, a practice which may also minimize participant burden.

While designed to be administered in its entirety, individual sections of the LLP user fall event survey may also be administered independently. Researchers or clinicians may administer specific sections (e.g., fall consequences) or questions (e.g., fall direction, impact location) to reduce respondent burden, and/or focus on areas of scientific or clinical interest. Should a targeted data collection be preferred, it is recommended that all original response options per question be retained. If specific response options are deemed unnecessary to the research question or clinical application, reintroduction of the open-ended "*Other*" response option in each question is recommended to ensure that relevant details of the fall event are not overlooked.

## 5.0 Conclusion

To improve the documentation of fall events in LLP users, we developed a novel structured LLP user-specific fall event survey (S2 Appendix) using best practices, including input from target respondents (i.e., focus groups and cognitive interviews with LLP users) [43, 49–51]. The resulting fall event survey is a 37-question instrument created to record and report fall frequency (i.e., number of events), fall circumstances (i.e., activity, surroundings, situation, and mechanics), and fall consequences (i.e., physical, financial, functional, behavioral, and psychological) in LLP users with a standardized and consistent approach. The LLP user fall event survey will help researchers and clinicians gather, document, track, and compare fall events among their participants and patients with greater ease, detail, consistency, and confidence. In doing so, the body of evidence required to design, test, and justify fall prevention initiatives to the individual needs of LLP users can be significantly improved.

## Supporting information

**S1 Appendix. Cognitive interview guides.**
(DOCX)

**S2 Appendix. Lower limb prosthesis user fall event survey.**
(DOCX)

**S1 Dataset. Data from preliminary validation study.**
(XLSX)

## Acknowledgments

The authors would like to acknowledge Janis Kim, Rana Salem, and Dana Wilkie for their assistance with recruitment, data collection, and data processing.

## Author Contributions

**Conceptualization:** Andrew Sawers, Brian J. Hafner.

**Data curation:** Andrew Sawers, Cody L. McDonald.

**Formal analysis:** Andrew Sawers, Cody L. McDonald.

**Funding acquisition:** Andrew Sawers, Brian J. Hafner.

**Investigation:** Andrew Sawers, Brian J. Hafner.

**Methodology:** Andrew Sawers, Brian J. Hafner.

Project administration: Andrew Sawers, Brian J. Hafner.

Resources: Andrew Sawers, Brian J. Hafner.

Software: Andrew Sawers.

Supervision: Andrew Sawers, Brian J. Hafner.

Validation: Andrew Sawers.

Visualization: Andrew Sawers.

Writing – original draft: Andrew Sawers.

Writing – review & editing: Andrew Sawers, Cody L. McDonald, Brian J. Hafner.

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
