## [Decision Letter · Decision Letter 0]

15 Jun 2022

PONE-D-22-10467A survey for characterizing details of fall events experienced by lower limb prosthesis usersPLOS ONE

Dear Dr. Sawers,

Thank you for submitting your manuscript to PLOS ONE. After careful consideration, we feel that it has merit but does not fully meet PLOS ONE’s publication criteria as it currently stands. Therefore, we invite you to submit a revised version of the manuscript that addresses the points raised during the review process.

We look forward to receiving your revised manuscript.

Kind regards,

Arezoo Eshraghi, Ph.D.

Academic Editor

PLOS ONE

Journal Requirements:

Reviewers' comments:

Reviewer's Responses to Questions

**Comments to the Author**

1. Is the manuscript technically sound, and do the data support the conclusions?

Reviewer #1: Yes

2. Has the statistical analysis been performed appropriately and rigorously? 

Reviewer #1: Yes

3. Have the authors made all data underlying the findings in their manuscript fully available?

Reviewer #1: Yes

4. Is the manuscript presented in an intelligible fashion and written in standard English?

Reviewer #1: Yes

5. Review Comments to the Author

Reviewer #1: This a much-needed study in this field of research and I am glad to see the authors solicited input from key stakeholders in survey development. The manuscript is well written and demonstrates scientific rigour.

Abstract

- Lines 40-41: this sentence is a bit vague – can you describe what the expected findings were? Or perhaps just remove the ‘i.e.’ bracket and describe this later

Introduction

- Lines 52-56: are these statistics focused on older adults with LLP or any age group?

- Please add a working definition of falls into this section for clarity

Methods

- Why was experiencing a fall (within a certain timeframe) not an inclusion criterion for the validation study? Were individuals allowed to complete the survey evenif they had never fallen? I am also interested to know when the ideal time to complete this survey would be following a fall – is it no longer valid after a certain timepoint?

- Please add the cognitive interview questions as an appendix for those who may be interested

- Line 169: Were the questions reviewed by the participants randomized or were the same 4-6 questions reviewed by all members of a particular group?

- Please add more details to data analysis section for validation study – for example, what was alpha set at?

Results

- Very comprehensive and well-written

Discussion

- Line 787: this paragraph is not well supported and unnecessary

- Line 817: Suggest removing this recommendation as the response time is brief and the survey has not been validated at the sub-category level

- Are there plans to validate this survey prospectively? Please describe your future directions.

Survey

- Are there any questions about which is the intact limb? This information would be useful when you ask things about the right and left legs (e.g., question 5)

- It is interesting that all emotions/changes listed in questions 34 and 35 are negative – do you think this impacts fear of falling in our patients?

6. PLOS authors have the option to publish the peer review history of their article (what does this mean?). If published, this will include your full peer review and any attached files.

Reviewer #1: No

---

## [Author Response · Author response to Decision Letter 0]

16 Jun 2022

We would like to thank the Academic Editor and reviewer for the detailed feedback and this opportunity to respond to their questions, comments, or concerns. Responses to each comment are provided below in italics. Revisions made to the manuscript are provided inline with each response, as appropriate. Within the revised manuscript, red underlined text indicates new text, while strikethrough indicates text that was deleted. All page and line numbers are associated with the marked-up version of the revision. A clean version of the revised manuscript (i.e., with revisions accepted) is also provided. Our minimal dataset is included within Supplement 2: Dataset. 

Reviewer #1 

1. This a much-needed study in this field of research and I am glad to see the authors solicited input from key stakeholders in survey development. The manuscript is well written and demonstrates scientific rigor. 

Response: Thank you. 

2. Abstract (Lines 40-41): this sentence is a bit vague – can you describe what the expected findings were? Or perhaps just remove the ‘i.e.,’ bracket and describe this later

Response: As suggested by the reviewer, we have removed this text from the abstract. The expected findings are described later in the manuscript (lines 193-195; 205-208). 

Revised Text: Administration of the fall survey to a national sample of 235 lower limb prosthesis users in a cross-sectional preliminary validation study, found survey questions to function as intended. Revisions to the survey were made at each stage of development based on analysis of participant feedback and data.

3. Introduction (Lines 52-56): are these statistics focused on older adults with LLP or any age group?

Response: The fall statistics cited in the introduction (lines 52-56) come from studies that included LLP users where the mean age varied between 48.7 to 62. Consequently, there is no explicit focus on older adults (i.e., > 65 years old) with lower limb amputation. The authors are only aware of a single study on fall event in LLP users that focused on older adults with lower limb amputation, Anderson et al., 2021. This study is not cited until later in the manuscript owing to its focus on the circumstances rather than incidence of falls in LLP users. 

Anderson CB, Miller MJ, Murray AM, et al. Falls after dysvascular transtibial amputation: A secondary analysis of falling characteristics and reduced physical performance. PM & R 2021; 13(1):19-29.

4. Introduction (Lines 52-56): Please add a working definition of falls into this section for clarity. 

Response: It is difficult to provide a straightforward and succinct definition of falls that would apply to all studies cited in lines 52-56. Two of the cited studies did not offer a definition at all, and the definitions used in the remaining seven studies are quite varied. As a result, any working definition of falls included here may not accurately reflect the specifics of each study and would likely require so much explanation that readability would be reduced. For these reasons, would respectively prefer not to include a working definition of a fall in this section. We discuss this problem, however, in the discussion section 4.2.

5. Methods: Why was experiencing a fall (within a certain timeframe) not an inclusion criterion for the validation study? Were individuals allowed to complete the survey even if they had never fallen? I am also interested to know when the ideal time to complete this survey would be following a fall – is it no longer valid after a certain timepoint?

Response: A fall or near-fall in the past 12 months was not an inclusion criteria for the preliminary validation study but was required to complete the fall survey (i.e., lines 542-544: 235 LLP users participated in the study, but only 212 recalled 1 or more falls or near-falls in the past 12 months and completed the fall event survey). While most of the fall survey is dedicated to documenting details of fall events, the first few pages serve to document whether a LLP user has fallen in the past 12 months, and if so, how many times. Therefore, even if a subject had not experienced a fall or near-fall in the past 12 months they would still have been able to complete the first part of the survey about their 12 month fall history. If fewer participants in the preliminary validation study had reported a fall or near-fall, we would have continued to recruit subjects until approximately 200 fall-related events were collected. 

6. Methods: Please add the cognitive interview questions as an appendix for those who may be interested. 

Response: We have included the questions for each of the cognitive interview groups as an appendix. 

Revised Text: An interview guide with a list of scripted open-ended questions (i.e., probes) was used to assess candidate survey questions and response options for clarity (i.e., was the intended purpose of the question clear), comprehension (i.e., was the question understood similarly across participants), and applicability (i.e., could the question be answered using the given response options) [70] (S1 Cognitive interview guides).

7. Methods (Line 169): Were the questions reviewed by the participants randomized or were the same 4-6 questions reviewed by all members of a particular group?

Response: Participants within each cognitive interview group of 5 LLP users answered probes about the same 4-6 survey questions (25-30 probes). However, questions presented to participants differed between groups. The order of the questions within each interview was randomized. Areas of the survey from which questions and probes were developed were mixed across interview groups (e.g., questions about fall consequences were distributed across cognitive interview groups rather than presented in just one group). Our approach was consistent with best practices (Willis, 2005) and our previous use of cognitive interviews (Morgan, 2014). We have added text to the methods to clarify this point for readers. 

Revised Text: Five interview guides consisting of scripted open-ended questions (i.e., probes) were used to assess candidate survey questions and their response options for clarity (i.e., was the intended purpose of the question clear), comprehension (i.e., was the question understood similarly across participants), and applicability (i.e., could the question be answered using the given response options) [70] (S1 Cognitive interview guides). Each interview guide included the same probes about survey instructions, fall event definitions, and fall history. Interview guides were limited to probing 4 to 6 of the survey questions to help participants remember how they arrived at their response, limit interview time, and ensure that each question was reviewed by at least 5 participants [62]. The 4-6 survey questions differed in each of the interview guides. The order of the questions within each interview was randomized. Areas of the survey from which questions and probes were developed were mixed across interview groups (e.g., questions about fall consequences were distributed across cognitive interview groups rather than presented in just one group). Interviews were audio-recorded and combined with field notes for subsequent analysis.

Willis GB. Cognitive interviewing : a tool for improving questionnaire design. Thousand Oaks, Calif.: Sage Publications, 2005.

Morgan SJ, Amtmann D, Abrahamson DC, et al. Use of cognitive interviews in the development of the PLUS-M item bank. Qual Life Res 2014; 23: 1767-1775.

8. Methods: Please add more details to data analysis section for validation study – for example, what was alpha set at?

Response: Much of the analysis for the preliminary validation study was descriptive, and thus there is little additional detail to be added. For those inferential tests that were run (i.e., Chi-square and McNemar tests), p-values have been added. 

Revised Text: To determine if survey questions operated as intended, Chi-square and McNemar tests were run to test for expected associations between fall circumstances and consequences (e.g., forward fall and impact with hands or knees), as well as expected patterns in survey responses (e.g., “problem with prosthesis” and “wearing prosthesis” were endorsed together). For statistical tests, � was set to .05. Statistical analyses were conducted using SPSS Statistics 28 software (IBM, Chicago, IL). The frequency of endorsement and content of participants’ responses to the open-ended “Other” response option included with each question in the preliminary validation study survey was analyzed to assess and enhance the range of fall circumstances and consequences included in the fall event survey.

9. Results: Very comprehensive and well-written.

Response: Thank you. 

10. Discussion (Line 787): this paragraph is not well supported and unnecessary. 

Response: We respectfully disagree with the reviewer. We believe that data from the administration of our fall circumstance and consequence survey may help identify what types of falls sensors can accurately detect, provide important context to falls data collected by wearable sensors (as noted by Ojeda, 2019), and inform revisions to fall detection algorithms so that wearable sensors can accurately detect falls that have the greatest impact on the lives of LLP users. We have however revised one sentence, as we cannot yet provide data or a citation to suggest that our fall survey would provide more consistent or specific details about a fall than voice recordings at or near the time of a fall.

Revised Text: As wearable sensors become more prevalent, and interest in their use to detect [92, 93] as well as understand the movements of real-world falls becomes of increasing interest [86, 94-96], the LLP user fall event survey is uniquely suited to provide the context required to interpret the voluminous and noisy data wearable sensor generate. In fact, descriptions of the circumstances surrounding falls using voice recorders were critical to interpretation of otherwise noisy data collected with inertial measurement units during fall events in older adults [97]. Administering the fall event survey in concert with the deployment of wearable sensors may help explore key movements during real-world fall events in LLP users. In doing so, the same data may contribute to concurrent improvements in the sensitivity and specificity of algorithms intended to provide automated detection of fall events in LLP users from wearable sensor data [92, 93, 98]. 

Ojeda LV, Adamczyk PG, Rebula JR, et al. Reconstruction of body motion during self-reported losses of balance in community-dwelling older adults. Med Eng Phys 2019; 64: 86-92.

11. Discussion (Line 817): Suggest removing this recommendation as the response time is brief and the survey has not been validated at the sub-category level. 

Response: We respectfully disagree with the reviewer. Validation at the sub-category level is not necessary for a descriptive instrument like the LLP user fall event survey. Validation of a sub-category or sub-scale is only required for scored self-report instruments, like the Prosthesis Evaluation Questionnaire (PEQ), that include more than one scale. Because results from a descriptive survey are reported at the item level, validation of the instrument as a whole is not necessary. Instead, efforts should focus assessing the quality of the individual questions. As noted in the manuscript, each question in our survey was found to be clear, well-understood, comprehensive, and operate as intended (i.e., responses were consistent with expected patterns and existing literature).Thus, users should feel comfortable using individual items or groups of items (e.g., sub-categories) without concern. While we agree that the overall survey is brief and is likely to be used in total, we anticipate that some researchers and clinicians may want to use only select sections of the survey (e.g., consequences). Therefore, we wanted to provide suggestions for best-practices if users elect to utilize the LLP user fall event survey in this manner.

12. Discussion: Are there plans to validate this survey prospectively? Please describe your future directions.

Response: There are plans to use the LLP user fall event survey in a prospective study and record the most recent rather than most significant fall. Fall events, as well as their circumstances and consequences, will be recorded as they happen and monitored longitudinally. We have included additional text in the discussion to highlight this as an area of future work. 

Revised Text: Identifying existing and establishing additional benefits of the LLP user fall event survey is an on-going area of research. A direct comparison between data collected with the fall event survey and a conventional patient interview would serve to clarify the “value added” of using a standardized fall survey over less structured approaches. The translation and validation of the LLP user fall event survey into other languages and cultures would also add to its value in the prosthetic community by increasing access to the survey internationally and ensuring valid and meaningful comparisons of fall events across cultures and languages. Prospective administration of the fall event survey (i.e., administering the survey immediately after falls occur) would serve to better characterize the range of fall events experienced by LLP users, and expand upon knowledge about the “most significant” fall events we measured in the preliminary validation study. Finally, additional research to expand and revise survey content to reflect fall experiences of other clinical populations (e.g., stroke, multiple sclerosis) would uniquely position researchers to compare important details of fall events between clinical populations and highlight areas of overlap as well population-specific needs with a single instrument. 

13. Survey: Are there any questions about which is the intact limb? This information would be useful when you ask things about the right and left legs (e.g., question 5). 

Response: We expect that survey users would obtain amputation-related information such as side of amputation using ad hoc questions as part of an intake survey. We did not include such content in the fall survey itself for this reason. With knowledge of the affected limb(s), investigators could then interpret results from the survey. We also choose to use the words “left” and “right” in the survey to accommodate respondents with unilateral or bilateral lower limb amputation. In the case of bilateral LLP users, terms such as “intact” and “amputated” would likely be perceived as confusing and/or exclusive. We have added text to the instructions of the survey to highlight this point for users of the fall survey. 

Revised Text: Note: Amputation-related information should be collected using ad hoc questions as part of an intake survey to assist with interpretation of fall survey results. Specifically, details regarding unilateral or bilateral amputation, the affected or amputated side if unilateral (i.e., left, or right), and the level of amputation on each side if bilateral.

14. Survey: It is interesting that all emotions/changes listed in questions 34 and 35 are negative. Do you think this impacts fear of falling in our patients?

Response: We had initially considered including positive emotions or changes such as “more confident in your balance”, “less afraid of falling”, but upon further discussion we did not consider these “positive outcomes” to be conceptually associated with fall-related consequences. Given that the nature of the section of the survey was to capture adverse outcomes, these “positive outcomes” were not included. 

Whether the historical emphasis on negative emotions and changes impacts fear of falling in LLP users remains to be determined, as does what types of falls may have a positive impact on balance confidence and fear of falling.

---

## [Decision Letter · Decision Letter 1]

13 Jul 2022

A survey for characterizing details of fall events experienced by lower limb prosthesis users

PONE-D-22-10467R1

Dear Dr. Sawers,

We’re pleased to inform you that your manuscript has been judged scientifically suitable for publication and will be formally accepted for publication once it meets all outstanding technical requirements.

Kind regards,

Arezoo Eshraghi, Ph.D.

Academic Editor

PLOS ONE

Additional Editor Comments (optional):

Reviewers' comments:

Reviewer's Responses to Questions

**Comments to the Author**

1. If the authors have adequately addressed your comments raised in a previous round of review and you feel that this manuscript is now acceptable for publication, you may indicate that here to bypass the “Comments to the Author” section, enter your conflict of interest statement in the “Confidential to Editor” section, and submit your "Accept" recommendation.

Reviewer #1: All comments have been addressed

2. Is the manuscript technically sound, and do the data support the conclusions?

Reviewer #1: (No Response)

3. Has the statistical analysis been performed appropriately and rigorously? 

Reviewer #1: (No Response)

4. Have the authors made all data underlying the findings in their manuscript fully available?

Reviewer #1: (No Response)

5. Is the manuscript presented in an intelligible fashion and written in standard English?

Reviewer #1: (No Response)

6. Review Comments to the Author

Reviewer #1: Thank you for your response, the additions have strengthened the manuscript. Including the cognitive interview questions has added great value.

7. PLOS authors have the option to publish the peer review history of their article (what does this mean?). If published, this will include your full peer review and any attached files.

Reviewer #1: No

---

## [Editor Report · Acceptance letter]

20 Jul 2022

PONE-D-22-10467R1 

A survey for characterizing details of fall events experienced by lower limb prosthesis users 

Dear Dr. Sawers:

I'm pleased to inform you that your manuscript has been deemed suitable for publication in PLOS ONE. Congratulations! Your manuscript is now with our production department. 

Kind regards, 

on behalf of

Dr. Arezoo Eshraghi 

Academic Editor

PLOS ONE